



# Observation-based sowing dates and cultivars significantly affect yield and irrigation for some crops in the Community Land Model (CLM5)

Sam S. Rabin[1,2], William J. Sacks[2], Danica L. Lombardozzi[2], Lili Xia[1], and Alan Robock[1]

[1]Department of Environmental Sciences, Rutgers University. 14 College Farm Rd., New Brunswick, New Jersey, 08901-8551, USA.
[2]Climate and Global Dynamics Laboratory, National Center for Atmospheric Research. 1850 Table Mesa Dr., Boulder, Colorado, 80305, USA.

**Correspondence:** Sam S. Rabin (sam.rabin@gmail.com)

**Abstract.** Farmers around the world time the planting of their crops to optimize growing season conditions and choose varieties that grow slowly enough to take advantage of the entire growing season while minimizing the risk of late-season kill. As climate changes, these strategies will be an important component of agricultural adaptation. Thus, it is critical that the global models used to project crop productivity under future conditions are able to realistically simulate growing season timing. This is

especially important for climate- and hydrosphere-coupled crop models, where the intra-annual timing of crop growth and management affects regional weather and water availability. We have improved the crop module of the Community Land Model (CLM) to allow the use of externally-specified crop planting dates and maturity requirements. In this way, CLM can use alternative algorithms for future crop calendars that are potentially more accurate and/or flexible than the built-in methods.

Using observation-derived planting and maturity inputs reduces bias in the mean simulated global yield of sugarcane and

cotton but increases bias for corn, wheat, and especially rice. These inputs also reduce simulated global irrigation demand by 15%, much of which is associated with particular regions of corn and rice cultivation. Finally, we discuss how our results suggest areas for improvement in CLM and, potentially, similar crop models.

## 1 Introduction

Over the coming decades, climate change will reduce the productive capacity of existing agricultural land, especially in warm

regions of the Global South such as Africa and South America (Kerr et al., 2022) (Hasegawa et al., 2022). Even under an optimistic climate change scenario, without adaptation, global production in currently-cultivated areas would fall below the recent historical mean by 2051 for maize and 2025 for wheat (Jägermeyr et al., 2021b). Adaptation measures—for example, shifting cultivated areas, increasing the use of fertilizer and irrigation, and improving crop cultivars—could mitigate productivity losses, helping the future global agricultural system meet the demands of an increasing and increasingly-affluent population

(Kerr et al., 2022). However, the particular forms, required magnitude, and potential benefit of adaptation—as well as stakeholders' capacity to adapt—will vary greatly by region, crop, and severity of climate change (Hoegh-Guldberg et al., 2022).





Some risk factors particular to crop production have to do with climate change impacts on plant phenology and management calendars. Shifts in the seasonal timing and level of temperature and rainfall, as well as the timing and likelihood of temperature and hydrological extremes, affect the optimal timing of management activities such as crop sowing. Likewise, warmer growing

seasons mean that crops mature faster, which all other things being equal can reduce yield because they have less time to photosynthesize. Such effects have already been observed in recent decades and can be expected to continue (Oort and Zwart, 2018; Zabel et al., 2021), but a wide variety of related management adaptations could help farmers reduce yield losses or even improve production. The potential of these adaptations has indeed already been demonstrated. Farmers in tropical and subtropical Asia have shifted sowing dates and adopted faster-maturing cultivars to minimize the risk of dangerous temperature

and hydrological extremes (Bahinipati et al., 2021; Shaffril et al., 2018; Shrestha et al., 2018; Wang et al., 2022). In temperate and cooler biomes, crop cultivars historically needed to mature quickly to deal with short growing (i.e., warm) seasons. As the climate has warmed and growing seasons have lengthened, planting earlier and using slower-maturing varieties has allowed farmers to avoid losses due to accelerated development, and even to increase yields (Kerr et al., 2022). In general, crop variety and calendar changes will continue to be important adaptation measures worldwide (Minoli et al., 2022), although the potential

of existing varieties is limited, especially under more extreme scenarios (Zabel et al., 2021).

Modeling provides a tool with which we can assess such impacts and adaptive interventions on various parts of the Earth system. Global crop models, such as those participating in the Global Gridded Crop Model Intercomparison (GGCMI; Jäger-meyr et al., 2021b), are particularly well-suited to exploring long-term productivity trajectories and the adaptive capacity of the agricultural system under different climate scenarios at large spatial scales. Intercomparison efforts such as GGCMI enable the

quantification of the uncertainty associated with not just climate scenario and model output, but also with crop model structural differences. GGCMI ensemble outputs have thus been used to explore a number of questions related not just to future crop yields (Müller et al., 2015; Jägermeyr et al., 2021b), but also irrigation water demand (Wada et al., 2013), the sustainability challenges facing food production across both land and sea (Blanchard et al., 2017), and food security after a hypothetical regional nuclear conflict (Jägermeyr et al., 2020).

GGCMI simulation protocols require participating models to be forced with a standardized, observation-based growing season dataset (Jägermeyr et al., 2021a, b). In part this is to avoid having different sowing dates be a potential confounding factor when comparing simulation outputs: Previous work has shown that correctly timing crop growing seasons is important for model performance with regard to yield (Dobor et al., 2016), and is sometimes even as important as simulating water stress (Jägermeyr and Frieler, 2018). However, cropland plays other important roles in the Earth system. The fluxes of matter

and energy associated with crop growth, fallow periods, use of irrigation and fertilizer, and other management can all affect local and regional weather at intra-annual time scales (Sacks et al., 2009). Likewise, irrigation needs and capacity can depend strongly on the timing of planting in regions where temperatures and/or water availability vary throughout the year. The accurate simulation of planting date and growing season length is thus especially important for crop models that are embedded in larger models of the land and Earth system.

One such model is the Community Land Model (CLM), a terrestrial system model that simulates a wide variety of processes including soils, hydrology, ecosystems, and land use. As part of the Community Earth System Model (CESM), CLM is capable





of exploring not just land processes but also how they interact with the rest of the Earth system. The crop module in the latest, fifth version of CLM (CLM5; Lawrence et al., 2019) performs well in terms of spatial distribution of and global total crop production for the recent past (Lombardozzi et al., 2020). However, some important crop growing seasons, such as Indian

wheat, are significantly mis-timed, which may contribute to underestimated yields when CLM tries to grow crops in suboptimal conditions (Lombardozzi et al., 2020). This could also cause incorrect timing of irrigation water demand, fertilizer application, and matter and energy fluxes—especially important for an atmosphere- and hydrosphere-coupled model such as CLM.

Here, we have given CLM the capability to use externally-prescribed growing season data such as that required by GGCMI, enabling for the first time its participation in this important community effort. We explore the effects of forcing CLM with

the GGCMI growing season data on simulations of historical yield and irrigation water demand, reinforcing the importance of simulating realistic seasons for crop model performance. Finally, we highlight where and how CLM performance could be improved.

## 2   Methods

### 2.1   Crops in CLM

The CLM crop module was adapted from the Agro-IBIS model (Kucharik and Brye, 2003) and first officially released as part of CLM4.5 (Levis et al., 2012; Lawrence et al., 2019). A large number of crops are parameterized as part of this module, but only six are included in default simulations (including this work): Corn, cotton, rice, soybean, sugarcane, and wheat. Corn and soybean are split into temperate and tropical varieties, and all wheat is represented by spring wheat due to ongoing development related to winter wheat. All crops can have both rainfed and irrigated area in each gridcell.

Each crop functional type (CFT; e.g., rainfed spring wheat) in CLM has a "sowing window" each year, which differs between the Northern and Southern Hemispheres. Planting occurs in this period if and when three CFT-specific thresholds are satisfied, all of which relate to temperature. This method allows sowing date to vary interannually with weather and to shift long-term with climate, but because it is purely driven by temperature, it can fail to properly represent growing seasons where water availability is more important in the sowing decision (Lombardozzi et al., 2020).

Maturity is determined in CLM not by time *per se*, but rather by "thermal time"—essentially, how warm the season has been for how long. Every day, the crop accumulates "growing degree-days" (GDDs; °C day$^{-1}$) equivalent to how far the day's mean temperature exceeds a CFT-specific "base temperature." Maturity occurs once the crop reaches its target GDD accumulation, $GDD_{mat}$, causing warmer-than-average growing seasons to be shorter than average (and vice versa). This method reflects temperature-dependent physiological processes associated with crop phenology and is widely used in crop modeling (e.g.,

Olin et al., 2015; Jägermeyr and Frieler, 2018). In CLM, $GDD_{mat}$ is calculated differently for various crop groups but is generally—within some minimum and/or maximum values—proportional to the 20-year running mean of spring and summer warmth. Henceforth, we will refer to $GDD_{mat}$ as "maturity requirements."





CLM usually harvests crops immediately upon reaching maturity. However, a maximum season length specified for each CFT sometimes causes premature harvest, which can affect the viability of the product (see Sect. 2.4). This maximum season

length prevents crops that failed to mature before the cold season from "overwintering" and resuming growth in the spring.

More details on the original formulation of crop calendars in CLM can be found in Appendix A1.

## 2.2 Externally-prescribed growing season criteria

Here, we have introduced code to allow gridded and optionally time-varying values of sowing date to be read from an input file, overriding sowing window and the related planting criteria. We have also enabled this functionality for $GDD_{mat}$.

The observation-based crop calendar dataset we use is provided by GGCMI as part of the phase 3 simulation protocol (Jägermeyr et al., 2021a, b). The six crops simulated here are included along with 15 others. This dataset derives typical sowing and maturity dates in each half-degree gridcell based on national governmental data where available, filling in other planted areas using a mix of existing global growing season products. For more details on the construction of the GGCMI crop calendar dataset, including data sources and gap-filling methods, see the supplementary material of Jägermeyr et al. (2021b).

We downscale the half-degree data from the GGCMI crop calendar dataset to the resolution of our simulations using nearest-neighbor remapping with the Climate Data Operators tool (Schulzweida, 2022). Sowing and maturity dates for two seasons of rice are provided for some gridcells; because CLM can only simulate one season per year, we choose the season associated with the greater area according to the maps provided in the dataset.

The GGCMI dataset only includes mean sowing and maturity dates, so some extra work is required to generate a $GDD_{mat}$

file for CLM to use. Specifically, we perform a "GDD-Generating" run for a baseline period (specified in the GGCMI protocol as the 1980–2009 growing seasons; Fig. 1), calculating the mean growing degree-days accumulated between the GGCMI sowing and maturity dates. (Where necessary, as specified by the GGCMI protocol, the maturity dates are moved forward so that CLM's maximum growing season length for each crop is not exceeded.) Those means are then mapped to generate a CLM input file for $GDD_{mat}$, with values specific to each CFT in each gridcell.

More details on this procedure can be found in Appendix A2.

## 2.3 Run setup

To test the code changes implemented in this work, we have performed a set of CLM runs at approximately 2-degree resolution (2.5° longitude × ~1.9° latitude, or the f19_g17 CLM grid) using climate forcings from the reanalysis-based GSWP3 dataset (Dirmeyer, 2011). While these runs do simulate crop yield, we do not necessarily expect improvements in crop yield perfor-

mance because so far we are only using static prescribed sowing date and $GDD_{mat}$ files. While CLM's native crop calendar algorithms are imperfect, that they allow variation over time could result in a performance advantage. However, these test runs may shed some light on CLM performance and suggest areas for improvement.

The model was spun up to equilibrium over 1802 years, forced with detrended 1901–1920 climate and no land use, using main-branch CLM code from the tag ctsm5.1.dev092. The next run segment continued with the same code and detrended

climate forcings but 1850–1900 land use from the CTSM5.2 dataset.

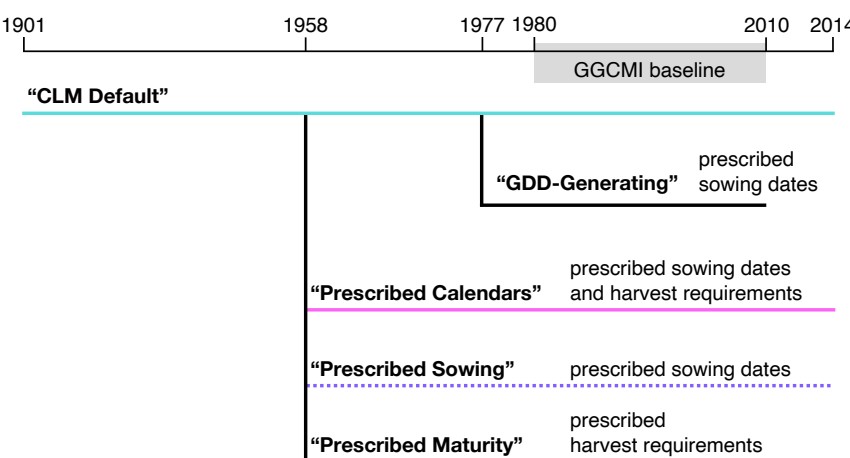

**Figure 1.** Runs performed for this work. Horizontal lines indicate model runs; a vertical line indicates the use of model state generated in the upper run used to initialize the lower run. Horizontal axis (simulated growing seasons) not to scale. Line colors and dashing reflect styles used for experiments in time series figures. Shading indicates the 1980–2009 growing seasons used to generate prescribed maturity requirements (following the GGCMI protocol) and analyzed here.

These spinup and land use initialization runs produced model states enabling historical-period runs to begin in 1901. These are described below and illustrated in Fig. 1.

The "CLM Default" run used our new code but with the normal CLM algorithms for determining sowing date and maturity requirements. The GDD-generating run branched off from CLM Default in 1977, producing maturity requirement maps as de-
scribed in Sect. 2.2 and Appendix A2. These, along with the GGCMI sowing dates, were then used in a "Prescribed Calendars" run branched from the CLM Default run at the beginning of 1958. To separately evaluate the effects of prescribed sowing dates and maturity requirements, we performed additional "Prescribed Sowing" (GGCMI3 sowing dates but CLM native maturity requirements) and "Prescribed Maturity" (vice versa) runs.

The start dates of the GDD-Generating and Prescribed runs were chosen because they are three years ahead of two important
years: 1961, the year in which the FAOSTAT yield data begin; and 1980, the first year in the GGCMI calibration period. Initial tests showed three years to be sufficient for dissipation of any explainable but unwanted behaviors associated with switching between model code versions and/or crop calendar settings.

Because CLM currently supports only one sowing of each CFT per calendar year in each gridcell, the practice of multi-cropping—producing two or more growing seasons of a crop per year on the same plot of land—is not possible. The GGCMI
growing season dataset includes two seasons for rice in some gridcells, of which we selected the season associated with the greater area.

The version of CLM used in these experiments is capable of supplying irrigation water from both surface and groundwater reservoirs. It can also limit the amount of irrigation withdrawals to 90% of main river channel volume in each gridcell, where



in the hydrogeographic dataset upon which CLM's river routing scheme is based, "main river courses" are defined as rivers fed
by at least 1000 3-arc-second gridcells ($\sim$8 km$^2$ at the Equator; Li et al., 2015; Lehner et al., 2022). However, these features are
still somewhat experimental, so our simulations allowed an unlimited supply of water for irrigation. We also used the default
irrigation scheme in which 100% of the water withdrawn for irrigation reaches the soil surface, where it can then infiltrate,
evaporate, or run off (Yao et al., 2022). Thus, plant irrigation demand, withdrawals, and application are all synonymous when
discussing our results.

## 2.4 Model evaluation

Because the GGCMI3 crop calendar dataset is designed for use in calibrating against the 1980–2009 growing seasons, we
restrict our analyses to that period. In figures, all maps are masked according to where each crop had at least one year with
nonzero area during the 1980–2009 calendar years.

### 2.4.1 Yield

We compare simulated yields with two historical yield datasets to understand the uncertainty in the observational data. The
FAOSTAT database (FAO, 2022a) provides country-level data on area harvested, production, and yield for every year starting
in 1961, which we aggregate to the global level for time series analyses. For spatial analyses, the EarthStat dataset provides
high-resolution (5-minute) maps of area harvested and yield for 175 crops in 2000 (Monfreda et al., 2008). We use a time-
varying version where the EarthStat data for 2000 are scaled to other years based on the relative difference for each country
in the FAOSTAT data between 2000 and the year in question (Lombardozzi et al., 2020). These are then aggregated to the
crops and spatial resolution used in our simulations. CLM harvest outputs are converted from tons carbon to tons biomass plus
moisture to match the observational data, as described in Appendix B.

In some analyses, we use maps of crop production (t) to identify the regions that contribute most strongly to global results.
Since EarthStat only provides yield and area, we must construct an EarthStat production dataset. The most straightforward
method would be to multiply the two EarthStat variables together, but this would complicate comparison with the CLM outputs,
since the EarthStat area maps differ from the maps used by CLM. Thus, we multiply EarthStat yields by CLM's crop areas to
produce a comparable observation-derived dataset.

To evaluate CLM's ability to reproduce general levels and trends in crop yield, we present a time series of global yield for
each of the six simulated crops, as well as total production of all crops and grain crops. These plots indicate the calculated
bias (mean across all years of simulated vs. observed yield, weighted by FAOSTAT production for each year) of the two
experimental simulations compared to FAOSTAT. To assess the spatial pattern of performance change, we also produce maps
of change in absolute bias relative to EarthStat, again weighted by (EarthStat) production in each year.

We also evaluate CLM's ability to capture interannual variability with Pearson's correlation coefficient ($r$), comparing each
year's simulated vs. observed yield. We again follow the example set by GGCMI (Müller et al., 2017). The data from each
time series is first detrended by subtracting from each point the mean of a five-year window centered on that point. This
removes bias and the effect of long-term trends due to, e.g., climate change or management improvements and allows a purer





evaluation of the model's ability to represent interannual variability. Again, FAOSTAT is used as the reference dataset. As in Müller et al. (2017), we test whether the correlation coefficients are significantly different from zero based on a two-sided t-test with $N-2$ degrees of freedom, where $N=30$ (growing seasons in 1980–2009). We also test whether the CLM Default

and Prescribed Calendars correlation coefficients differ significantly from each other using Fisher's Z transformation test for correlations (FAO, 2022b).

Comparing simulated and observed yields is not straightforward for several reasons, one of which regards how simulated crop "maturity" is determined. In CLM, crops can be harvested before reaching full maturity if a growing season is too cool and the maximum season length is reached. However, excluding all harvests at less than 100% maturity (as measured by

accumulated growing degree-days relative to the maturity requirement) would be unrealistic. The FAOSTAT standard is to track all crops that are either consumed by the producers or sold at market (FAO, 2022b), and crops can be viable for those purposes without reaching complete maturity. We adopt the standard used for GGCMI phase 3 (Jägermeyr et al., 2021b), which specifies a threshold of 90% maturity for all crops except corn, which has an 80% threshold to reflect its use as feed.

Another challenge to yield comparison is the potential for mismatches in how harvested crops are associated with calendar

years. CLM can save harvested yield either annually (i.e., the total amount harvested in a calendar year) or by growing season (i.e., the amount harvested from growing seasons that began in a calendar year). Agricultural statistics are subject to similar decisions: The FAOSTAT standard is to "refer production data to that [calendar] year into which the bulk of the production falls" if "it is not possible to allocate the relative production to" two consecutive years (FAO, 2022b), but there is a certain amount of ambiguity in that instruction and uncertainty in how well it is followed. Differing standards between observations and model

outputs for associating harvests with a year can suggest misleadingly poor model performance if not properly accounted for (Iizumi et al., 2021). As such, the standard GGCMI time series analysis method is to use growing season yields (i.e., associated with year of planting), shifting them forward or backward a year if doing so improves the Pearson's correlation coefficient by 0.3 or more (Müller et al., 2017). We follow this standard here for our time series analyses, using FAOSTAT as the reference dataset. Superscripts [L] and [R] in figure subplot titles indicate, respectively, leftward (where, e.g., simulated growing season 2000

is compared against observed growing season 1999) and rightward shifts. In figures, "Total" plots have shifts calculated and applied after summing the unshifted time series of their constituent crops. These shifts are only applied when examining global time series, not time-averaged maps.

Where crop model outputs (two variables, with time axes of growing seasons and calendar years) must be cross-referenced with land use areas (with time axis of calendar years), as in the calculation of global average yields (t ha$^{-1}$) or per-gridcell

production (t), we use the calendar-year outputs. So, for example, production for one gridcell in 1986 could include harvest from crops planted in 1985 and/or 1986.

### 2.4.2 Irrigation

To evaluate the influence of prescribed crop calendars on CLM's ability to simulate global irrigation, we produce time series figures of irrigation volume for the area associated with the six explicitly-simulated crops as well as for all land (including

other crops not explicitly simulated). The latter is compared to global estimates of global irrigation withdrawals, which unlike





the data from CLM generally include water that is lost between withdrawal and application. We also produce maps of change in mean annual irrigation demand for the six explicitly-simulated crops in order to understand which regions are most affected.

However, mean demand is only part of the story; it also matters when in the year irrigation is needed and how much is needed relative to supply (Hanasaki et al., 2008). Thus, we produce maps of the shift in timing of peak monthly irrigation demand as well as the change in maximum monthly irrigation use as a fraction of supply. As groundwater-sourced irrigation is not enabled in these simulations, we define "supply" as the volume of the main river channel in CLM (Sect. 2.3).

## 3 Results

### 3.1 Seasonality

Mean sowing dates over the 1980–2009 growing seasons for most crops in the CLM Default run differ notably from the GGCMI3 prescribed sowing date, with at least some regions differing by 165 days or more (where 182.5 days is the maximum possible difference in a 365-day year). Figure 2 illustrates this for spring wheat (chosen because it is widely planted in CLM, although high-latitude areas would be planted with winter wheat were that available in the model). As expected, the mean sowing dates in the Prescribed Calendars run match the GGCMI3 dates exactly (not shown). For most non-tropical crops, mean CLM Default sowing dates are closest to the observation-derived values in temperate regions, especially in North America and Europe (Figs. S1–7). This is consistent with the focus on such regions in the initial development and evaluation of Agro-IBIS (Kucharik, 2003; Kucharik and Brye, 2003) and when it was brought in to CLM (Levis et al., 2012). Temperate soybean and (to a lesser extent) temperate corn are the exceptions due to their not including tropical regions, but note that the tropical versions of those crops do show wide variation in sowing date performance.

Average growing season length for spring wheat in the CLM Default run is in many regions more than 30 days shorter than the GGCMI3 values, and in some places is more than 90 days shorter (Fig. 3). In contrast, the Prescribed Calendars run mostly produces growing season lengths within 30 days of the GGCMI3 values, except for regions where the GGCMI3 growing season exceeds the CLM maximum (for spring wheat: 150 days), which we retained when generating maturity requirements (Sect. 2.2 and Appendix A2). The results are similar for most other crops, although sugarcane seasons in both CLM Default and Prescribed Calendars are often more than 150 days shorter than GGCMI3 (which has 365-day growing seasons in most of the world, compared to the CLM maximum of 300), and rice has large areas where the CLM Default average season is 30–90 days longer than GGCMI3. As with sowing dates, growing season lengths for most crops are closest to GGCMI3 in temperate regions, with soybean being the exception (Fig. S8–15).

The GGCMI3-derived maturity requirements are much more geographically variable than the CLM Default values, as illustrated for rainfed spring wheat in Fig. 4. This crop is representative of most others in that (a) CLM Default values are closer to the GGCMI3-derived values in temperate regions than in the tropics, (b) a large portion of the planted area in the CLM original configuration uses the maximum possible GDD value from the CLM algorithm, (c) the GGCMI3-derived values exceed that in many places (Figs. S16–20). The latter two pieces of evidence together suggest that the maximum maturity requirement values in CLM are making it so that seasons are shorter than they should be for most crops, at least in terms of "thermal time"



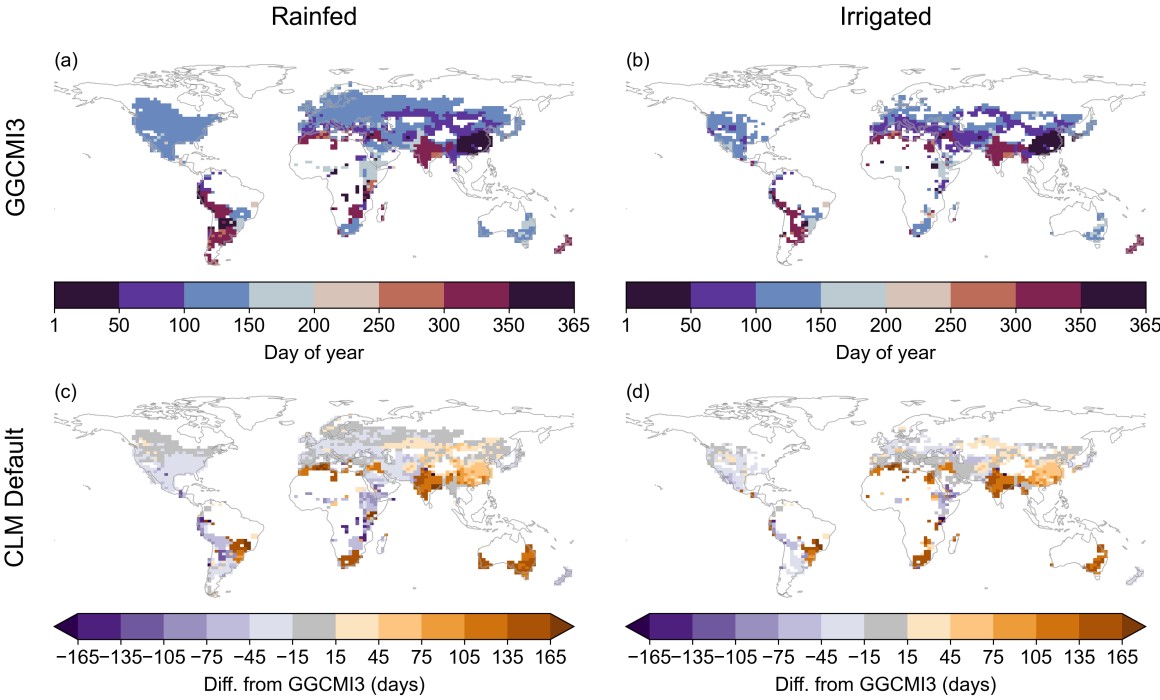

**Figure 2.** Sowing date for spring wheat in GGCMI3 prescribed calendars (a–b); differences in mean 1980–2009 sowing date from prescribed value for CLM Default run (c–d). Prescribed Calendars dates match GGCMI3 exactly and are thus not shown. Other crops can be found in Supplemental Figs. S1–7.

(Sect. 2.1). (In terms of days, a higher maturity requirement with the same sowing date would mean a longer season, but with a sowing date later in the spring might mean a shorter season.) Sugarcane and, to a lesser extent, temperate soybean are the exceptions to this pattern, with more centrally-balanced distributions of CLM Default maturity requirements.

## 3.2 Yield

### 3.2.1 Trend, bias, and spatial patterns

Using the GGCMI3-derived calendars affects yield performance differently for different crops (Fig. 5).

Neither the bias nor the trend of corn differs much between the CLM Default and Prescribed Calendars runs. Soybean mean bias is also similar between the runs, although the magnitude of interannual variability in the second half of the analyzed period seems to have improved in the Prescribed Calendars run while the reproduction of the observed trend worsened. A 0.6 t ha$^{-1}$ (21%) overestimate for wheat in CLM Default is increased in Prescribed Calendars to 1 t ha$^{-1}$ (50%).

Cotton, rice, and especially sugarcane see large increases in global yield in Prescribed Calendars relative to CLM Default. Strong relative underestimates for cotton and sugarcane are greatly improved, with the former's 0.5 t ha$^{-1}$ (33%) underesti-



**Figure 3.** Growing season length for spring wheat in GGCMI3 prescribed calendars (top row); differences in mean 1980–2009 sowing date from prescribed value for CLM Default run and Prescribed Calendars run (middle and bottom rows, respectively). Other crops can be found in Supplemental Figs. S9–15.

mate being almost completely resolved, and the latter's bias being reduced by nearly 25% (12.6 t ha$^{-1}$ underestimate to 9.6 t ha$^{-1}$ overestimate). Rice, on the other hand, goes from good performance (underestimate of only 0.3 t ha$^{-1}$) to a significant overestimate (by > 2 t ha$^{-1}$).

Due to the large shifts in global mean cotton, rice, and sugarcane yields, we map their mean differences in production and absolute bias between the CLM Default and Prescribed Calendars runs in Fig. 6. Figure S21 shows equivalent maps for corn, soybean, and wheat, which are not discussed here.

255



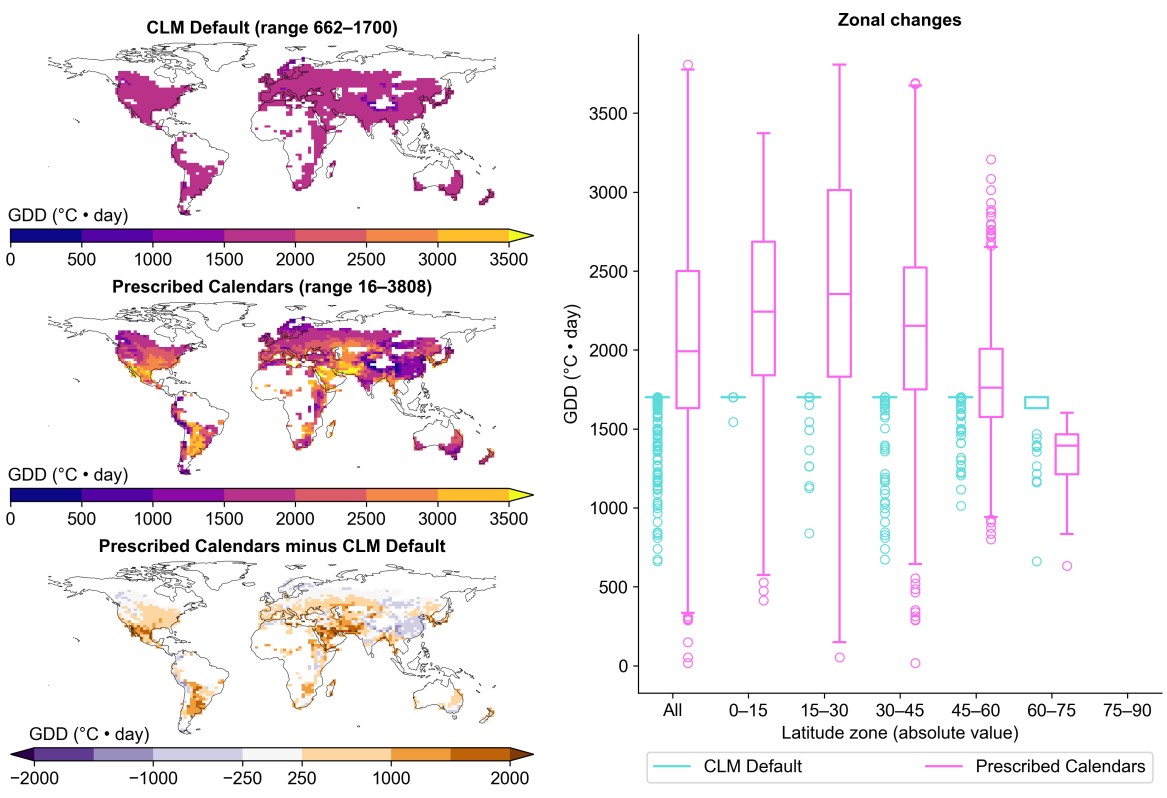

**Figure 4.** Maturity requirements for spring wheat (area-weighted average of rainfed and irrigated). Box plots compare, at different latitude zones, distributions of maturity requirements for CLM Default (cyan) and Prescribed Calendars (pink). Boxes' central lines are medians, box edges are 25th and 75th percentiles, and whiskers extend to the minimum and maximum of the data excluding any outliers (points outside the median ± 1.5 times the interquartile range; circles). Other crops can be found in Supplemental Figs. S16–20.

Cotton's increase in production when moving from CLM Default to Prescribed Calendars is driven mainly by India and China, with smaller contributions from Pakistan, the United States, and Uzbekistan (Fig. 6a; Fig. S22). This represents a substantial yield performance improvement in China, Uzbekistan, and to a lesser extent the United States (Fig. S23). However, 260 many of the gridcells with strong production underestimates (Fig. S24b) did not see much improvement. Thus, while CLM Default's cotton yields were too low on average, the Prescribed Calendars caused yield to increase too much in the wrong places (Fig. 6b).

The global increase in rice yield is driven mostly by increased production in China, India, and Thailand (Fig. 6c, S22). China's increase results in performance improvement there (Fig. S23), whereas India and Thailand become strongly overes-265 timated (Fig. 6d). The increase in sugarcane yield is mostly associated with India, Brazil, and Cuba (Fig. 6e, S22). Of these, performance in India is strongly improved, Brazil is slightly improved, and Cuba is worsened (Fig. 6f; Fig. S23).



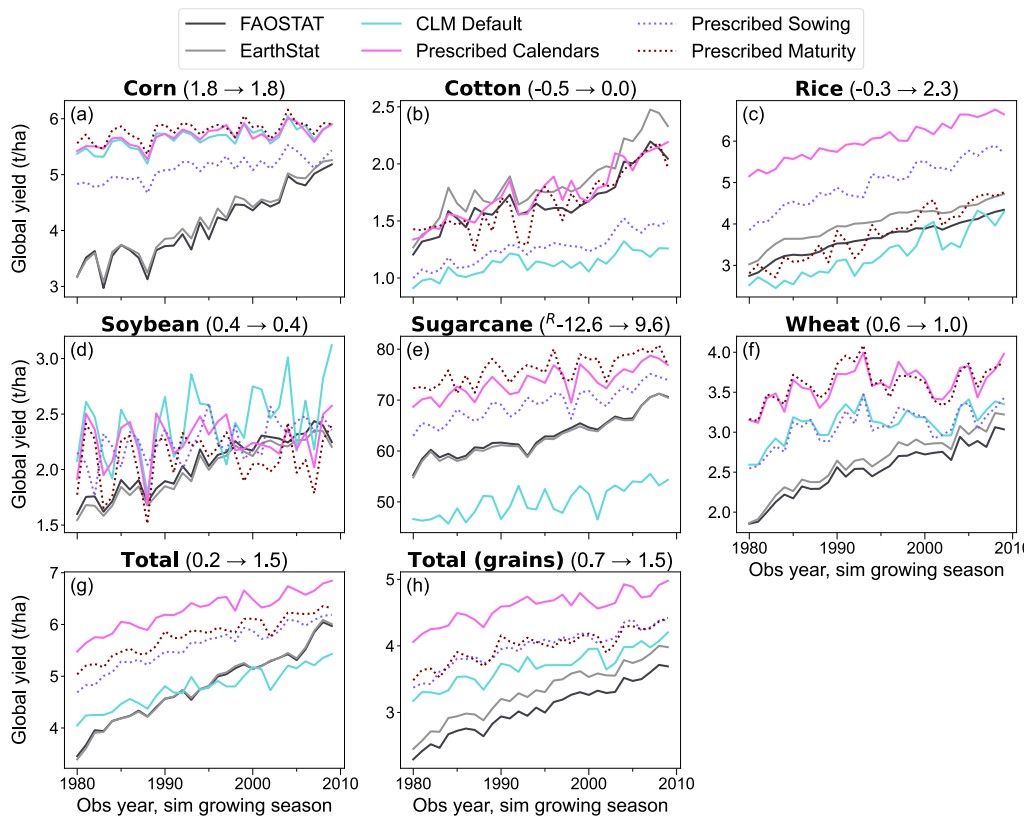

**Figure 5.** Observed (black and gray) vs. simulated (colors) global crop yield for 1980–2009. Numbers in parentheses are bias over that period of model outputs (CLM Default → Prescribed Calendars) in t ha$^{-1}$ relative to FAOSTAT, with superscripts denoting leftward or rightward shifts of simulations (see Sect. 2.4). "Total" is the production of all six crops combined divided by their area, with "grains" including all crops except cotton and sugarcane.

### 3.2.2 Attributing yield differences

Examining Fig. 5 for the Prescribed Sowing and Maturity runs reveals that the yield increases for cotton, rice, and sugarcane result do not occur for the same reasons. Rice sees little difference from CLM Default—at least at the globally-aggregated level—when using only the GGCMI3-derived maturity requirements, yield increases with only the GGCMI3 sowing dates, and a positive synergistic effect of using both. Sugarcane, on the other hand, actually sees an even stronger overestimate of yield in the Prescribed Maturity run, with the addition of GGCMI3 sowing dates bringing yield closer to observations. Cotton sees nearly all of its yield increase due to the Prescribed Maturity inputs, with the Prescribed Sowing inputs contributing a small further increase.

Rice has wide areas, including much of China, where it rarely or never reaches maturity in the CLM Default simulation (Fig. 7). This is alleviated in the Prescribed Calendars run due to the combination of new sowing dates and season lengths,







**Figure 6.** Spatial distribution of (a, c, e) difference in production and (b, d, f) difference in absolute production bias relative to EarthStat (see Sect. 2.4.1). Differences are calculated as the mean of Prescribed Calendars minus CLM Default over the 1980–2009 calendar years. This figure includes cotton (a–b), rice (c–d), and sugarcane (e–f); see Fig. S21 for equivalent figure with corn, soybean, and wheat. Cells with no planted area of each crop in CLM are masked (white) in (a, c, e); those plus cells with no EarthStat area are masked from (b, d, f). Gray cells are those outside the top 95% of cumulative absolute values in each map.

causing the performance improvement seen across China (Fig. 6d). The parts of India and Southeast Asia where yield increases resulted in worsened bias (Fig. 6c, d) never or rarely had trouble reaching maturity in the CLM Default run (Fig. 7). Under the CLM Default setup, rice in temperate regions is generally planted at the end of the sowing window (Fig. S25), which in the Northern Hemisphere is Feb. 28. Across much of the temperate zone it is too cool for a season started then to succeed in



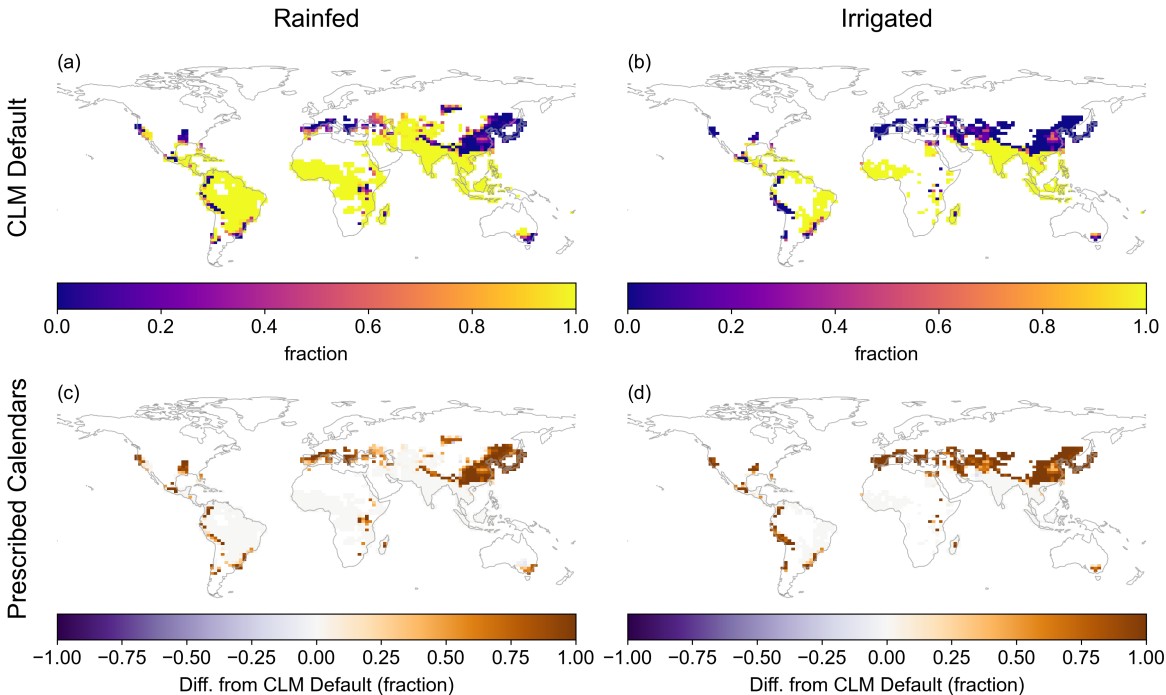

**Figure 7.** Top row: Fraction of rice harvests in the 1980–2009 growing seasons that were harvested at maturity (according to the 90% threshold; see Sect. 2.4). Bottom row: Change in fraction of rice crops harvested at maturity when moving to Prescribed Calendars.

most years, even with the reduced maturity requirements derived from the GGCMI3 crop calendar (Fig. S18). The Prescribed Sowing dates are also needed, which across the temperate region shift planting to the spring or early summer.

### 3.2.3 Global interannual variability

Using the GGCMI3-derived calendars seemingly has little effect on CLM's ability to reproduce observed interannual variability, as no change in correlation coefficient ($r$) is significant at the p < 0.1 level (Fig. 8). However, our 30 data points leave Fisher's Z transformation test somewhat underpowered, so here we discuss the differences observed.

Wheat's correlation increases slightly and corn's decreases slightly, but both remain significant at the p < 0.05 level. The correlation coefficients of the rest of the crops remain not significantly different from zero, although seemingly large changes are seen in some. For example, soybean's $r$ more than triples to 0.303, and sugarcane's is reduced by 97%. Rice's is reduced by more than half in the Prescribed Calendars run, although its amount of interannual variability is closer to the observed than is CLM Default's. For the grain crops (corn, rice, soybean, and wheat) as a whole, $r$ increases by 36%, from 0.442 (p < 0.05) to 0.600 (p < 0.01). The correlation coefficient for all six crops combined is only 0.240 in the Prescribed Calendars run, but this is an improvement from the CLM Default run's -0.001. Again, however, none of these changes are statistically significant.



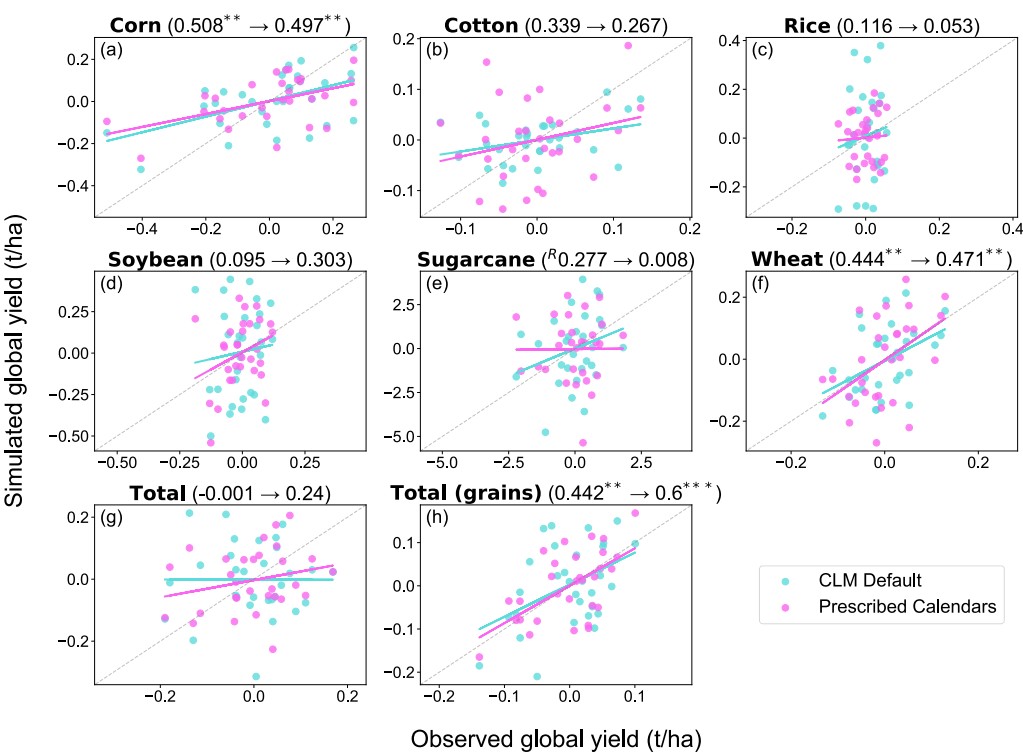

**Figure 8.** Scatter plots of observed (x axes) vs. simulated (y axes) global crop yield for 1980–2009 after detrending as described in Sect. 2.4. Numbers in parentheses are Pearson's correlation coefficient ($r$) of each simulation (CLM Default → Prescribed Calendars), with superscripts denoting leftward or rightward shifts of simulations (see Sect. 2.4). Asterisks indicate correlation coefficients that differ from zero at the $p < 0.1$ (*), $p < 0.05$ (**), or $p < 0.001$ (***) levels. Solid lines are best-fit lines, with dashed lines indicating the 1:1 line. See caption of Fig. 6 for definitions of "Total" and "Total (grains)."

The Prescribed Calendars correlation coefficients for corn, soybean, and wheat in Fig. 8 are on the low end of performance for models evaluated under the "fully harmonized" setup in GGCMI phase 1 by Müller et al. (2017); for rice, it performs worse than any model there. Many of the models in GGCMI phase 1 included effects of heat, cold, drought, and/or soil saturation to represent crop damage or death from extreme events; CLM's lack of representation of these processes likely affects its performance. Some of the GGCMI models also contain more crop-specific parameterizations of development and phenology that may provide a performance advantage. That said, our results are not directly comparable with the analyses in that paper: GGCMI phase 1 used different climate forcing and fertilizer input, and Müller et al. (2017) used a different crop area dataset for aggregating to global average yields. In addition, the coarser resolution of the CLM runs performed here (2.5° longitude × ~1.9° latitude) may smooth out finer-scale variation in growing season conditions that were captured in the half-degree runs performed for all but two of the models in Müller et al. (2017).



The poor performance seen for cotton and sugarcane is unfortunate but perhaps not surprising, as their physiology is very
different from the herbaceous grain crops for which the CLM crop module is mostly targeted. Indeed, for this reason cotton
and sugarcane are commonly not included in crop modules of dynamic global vegetation models such as CLM, and they are
not evaluated in Müller et al. (2017). Cotton, for example, is a shrub that is mostly now planted as an annual but was bred
from a perennial ancestor. Its indeterminate growth pattern contrasts with the determinate (and herbaceous) grain crops and
means that accumulated growing-season temperature alone is not a good predictor of development (Jans et al., 2021). Cotton
in the LPJmL model represents cotton plants as small trees and determines phenological status based on solar radiation, water
availability, and heat and cold stress (Jans et al., 2021). That implementation achieved a global correlation coefficient over
1980–2010 of 0.414, significant at the $p < 0.05$ level. CLM, in contrast, effectively represents cotton as another herbaceous
crop whose determinate growth is governed solely by accumulated heat units. The use by Jans et al. (2021) of country-specific
planting densities and half-degree resolution likely contribute to LPJmL's better performance, but CLM would likely benefit
from a more realistic cotton. Sugarcane also takes an indeterminate growth form which one global model, ORCHIDEE-STICS,
approximates using a special stress term (Valade et al., 2014). LPJmL and CLM do not account for this characteristic and have
both previously been shown to perform poorly at the global scale (Yin et al., 2023). In addition, CLM's performance with
regard to sugarcane is likely hampered by its inability to simulate sugarcane growing seasons longer than 300 days, when real
sugarcane is often harvested more than a year after planting (Jägermeyr et al., 2021b).

### 3.2.4   Yield sensitivity to viability threshold

The yield of some crops is highly dependent on the choice of minimum maturity level (heat unit index) for harvested biomass to
be included in yields. This is illustrated in Fig. 9, which compares the GGCMI3-based thresholds used here (80% maturity for
corn, 90% for other crops; Sect. 2.4) to thresholds of 100% and 0%. The CLM Default simulation for rice, for example, would
have seen a slightly worse underestimate in global mean yield with a 100% maturity threshold and a significant overestimate
with a 0% threshold. Similarly, the Prescribed Calendars simulation's improved cotton yield would be completely negated if
requiring 100% maturity.

### 3.3   Irrigation demand

The Prescribed Calendars run used ~15% less water (874 km$^3$ yr$^{-1}$) for irrigation than CLM Default (1023 km$^3$ yr$^{-1}$) over
1980–2009 when considering all cropland (Fig. 10h). Because the current version of CLM has been shown to underesti-
mate global irrigation demand (Yao et al., 2022) compared to estimates that are mostly in the range of 2,000–3,000 km$^3$ yr$^{-1}$
(Hanasaki et al., 2008; Wisser et al., 2008), to some extent this represents a degradation in performance. However, the sim-
plified representative irrigation behavior simulated by CLM does not fully encompass the variety of irrigation practices used
around the world. In particular, CLM does not include paddy irrigation, likely a key reason that global irrigation demand is
underestimated. New CLM parameterizations for sprinkler, flood, and paddy irrigation styles developed by Yao et al. (2022) re-
sulted in a more realistic simulation of global irrigation demand that was nonetheless too high by about 20–30%. (Indeed, those



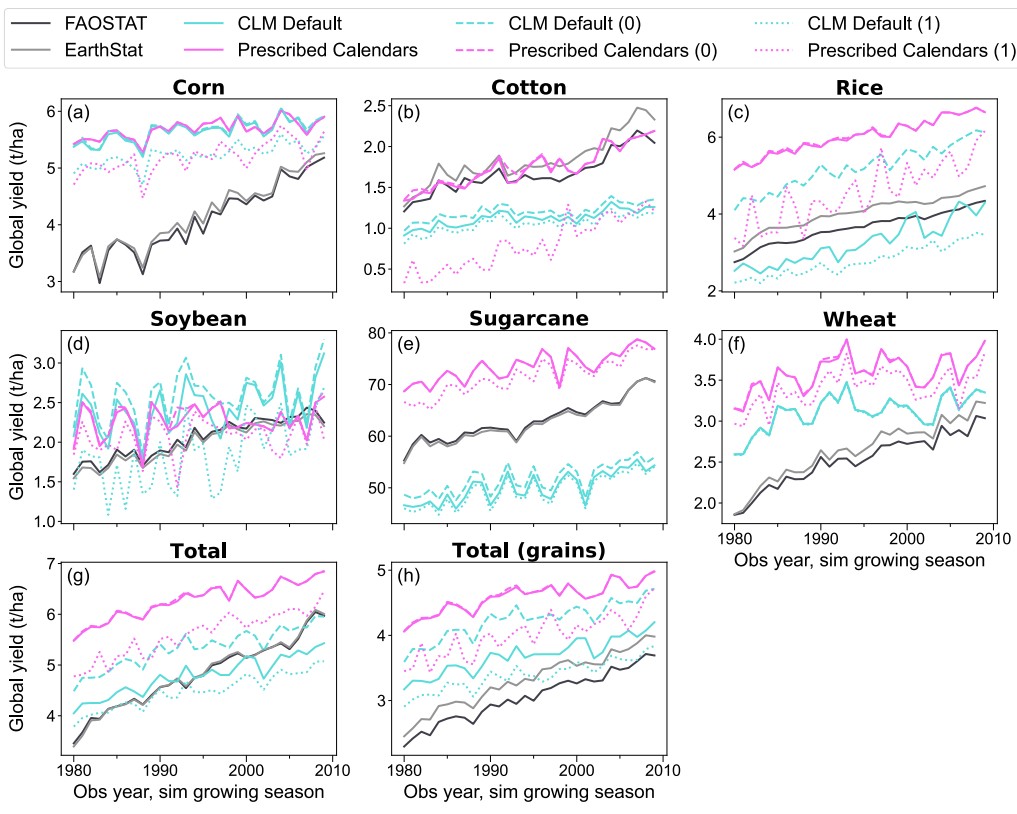

**Figure 9.** As Fig. 5, but with Prescribed Calendars and CLM Default runs under the GGCMI3 maturity thresholds (as Fig. 5; solid colored lines; Sect. 2.4) compared to the same runs with minimum heat unit index 0 (i.e., including all harvested biomass regardless of maturity level; dashed lines) and 1 (i.e., including only fully mature yields; dotted lines). Unlike Fig. 5, no timeseries shifts are included here, ensuring consistent year-to-year comparisons among the simulation data.

authors note that using observation-based crop calendars could be an important next step in refining irrigation in CLM.) Using that new irrigation system, then, the reduced demand simulated with the Prescribed Calendars could represent an improvement.

For the main six crops on which we have focused our analyses, mean irrigation use was ∼20% (∼101 km$^3$) lower under Prescribed Calendars (Fig. 10g). Of those, rice contributes the majority of the reduction (–100 km$^3$ yr$^{-1}$, –45%), with corn

(–15 km$^3$ yr$^{-1}$, –28%) and soybean (–7.6 km$^3$ yr$^{-1}$, –28%) seeing notable fractional reductions that contribute relatively less to the six-crop total irrigation demand. Cotton uses 30% more irrigation water (+18 km$^3$ yr$^{-1}$) under Prescribed Calendars, while sugarcane and wheat use only slightly more (< +2 km$^3$ yr$^{-1}$, < +7%).

For crops that saw reduced irrigation demand, the new sowing dates are mostly responsible, with the new maturity requirements having relatively little effect (Fig. 10). For example, Prescribed Calendars requires much less irrigation water for rice in

India and Bangladesh because it can be planted near the summer monsoon, rather than being restricted to sowing in January and February as in CLM Default (Fig. S25). Among the crops seeing increased demand, cotton and sugarcane see the opposite



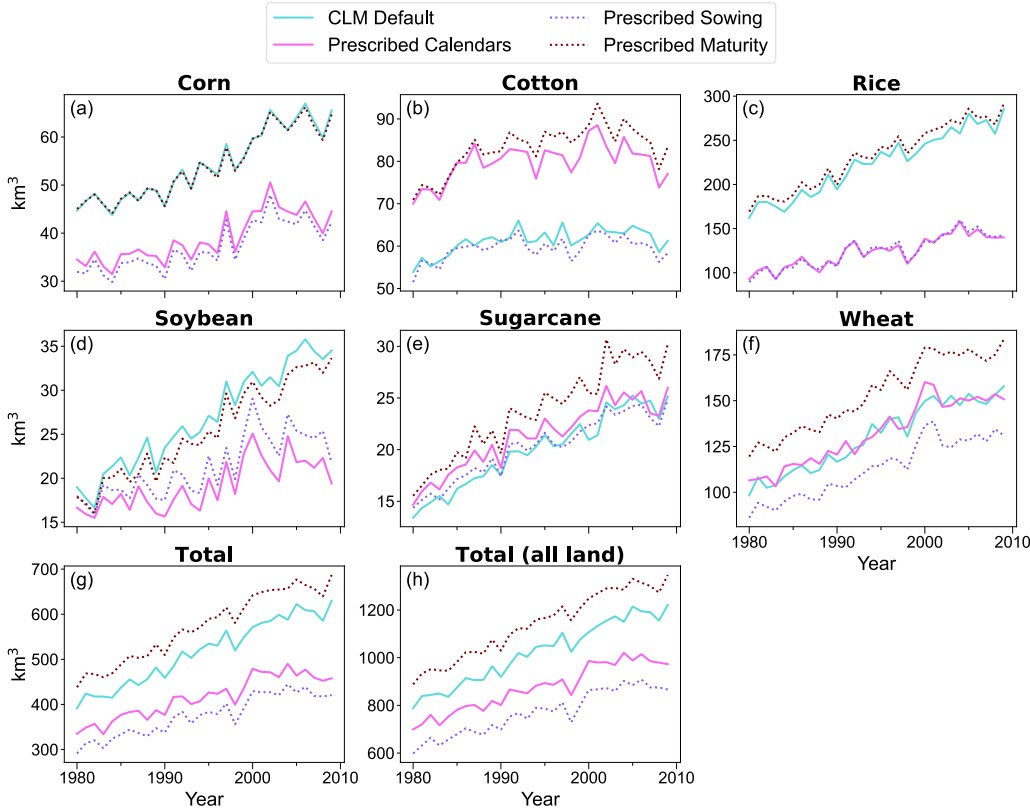

**Figure 10.** Time series of annual global irrigation use on the six main crop types analyzed here (a–g) as well as all land including other crops (h).

pattern—their often-extended growing seasons under the observation-derived calendars (Figs. S11, S15) require longer periods of irrigation and thus more water use overall. Wheat, in contrast, experiences a notable increase in demand due to the prescribed maturity requirements that is mostly counterbalanced by a decrease in demand associated with the prescribed sowing dates.

Despite the magnitude of the global differences evident in Fig. 10, one or two regions tend to be responsible for the effects of Prescribed Calendars on irrigation use for each crop. South Asia stands out as a major driver for all observed differences (Fig. 11), reinforcing the importance of getting the growing seasons correct there (Lombardozzi et al., 2020). Specifically, India and Bangladesh see most of the reduction in irrigation requirement (Figs. 11, S26a), mainly due to rice and to a lesser extent soybean and wheat (Fig. S27). Pakistan sees the strongest increases in irrigation (Figs. 11, S26b) and is a major contributor

to the global increases seen in cotton, sugarcane, and wheat (Fig. S27). One other important region is in China, where the agricultural areas to the south of Beijing contribute strongly to the corn decrease and cotton increase (Figs. S26, S27).

While the total amount of irrigation applied over a year is important, the timing of irrigation withdrawals has implications for how much water is left to sustain aquatic ecosystems and for other human uses, as well as for climate effects at regional





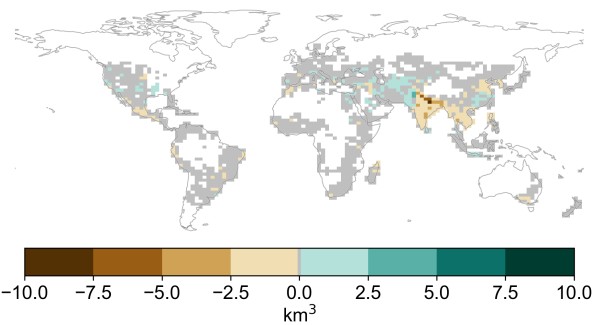

**Figure 11.** Global distribution of difference in mean annual irrigation demand, summed across the six crops analyzed here. Gray cells are those outside the top 95% of cumulative absolute values; white cells had no irrigated crops. SSee Fig. S26 for positive and negative components in each gridcell and Fig. S27 for maps of individual crops.

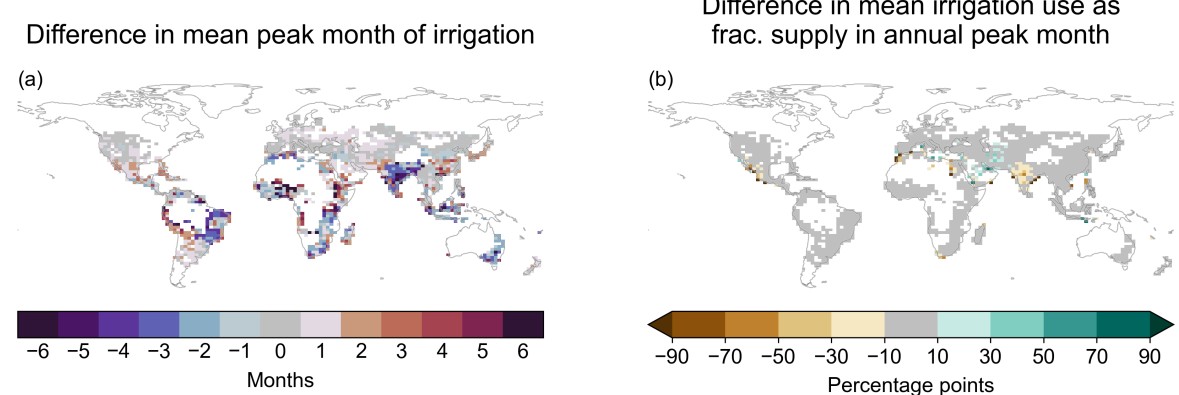

**Figure 12.** Mean 1980–2010 difference between CLM Default and Prescribed Calendars in (a) mean month of peak irrigation withdrawals and (b) maximum monthly irrigation use as fraction of main river channel volume (Sect. 2.3). In (a), white gridcells had no irrigation in either or both runs. White areas in (b) include those plus gridcells with no main river channel volume in month of maximum irrigation use.

and sub-annual scales. India and Bangladesh again stand out, with the shifted growing seasons over most of their area causing
withdrawals to peak three or more months earlier under Prescribed Calendars than CLM Default (Fig. 12a). The combination
of this seasonal shift with the previously-shown overall reduction in irrigation needs means that peak monthly irrigation usage,
when expressed as a fraction of overall river volume, decreases 10 percentage points or more across much of India (Fig. 12b).



## 4 Discussion

### 4.1 Multiple cropping

As mentioned in Sect. 2.3, CLM does not yet support the planting of multiple seasons of a crop (or different crops) in a single calendar year. Multicropping accounted for 12% of global crop area—including 10% and 13% of corn and wheat area, respectively—around the year 2000 (Waha et al., 2020). For regions where multiple cropping is common, the seasonal shifts observed when moving from the CLM Default setup to the Prescribed Calendars may represent a shift not from an unrealistic season to a realistic one, but from one actual season to another.

This effect is likely especially strong for rice, more than one-third the area of which is under multiple cropping (Waha et al., 2020). In particular, the area of northeast India responsible for much of the global reduction in simulated irrigation demand for rice (Fig. S27c) mostly sees two rice seasons per year (Frolking et al., 2006). It is thus plausible that the CLM Default setup represented the actual timing of the primary irrigated rice season, in which case the reduction in irrigation demand seen when moving to Prescribed Calendars would simply be an artifact of CLM's single-season limitation. (Of course, if simulating both 375 seasons under either CLM Default or Prescribed Calendars, the total demand would be higher than when simulating just one season.) The GGCMI3 crop calendars only include one rice season for most of the area in question, so a different observational dataset would be needed to explore this possibility.

### 4.2 Implications for CLM5 crop modeling

The results presented here suggest that the benefits of reworking growing seasons to be closer to those in the Prescribed 380 Calendars run would depend on the crop and metric of interest. Soybean is the most consistently improved, with its mean global yield not changing much but performance strongly improving in terms of global interannual yield variability (Fig. 8d) and spatial pattern of production (Fig. S21d). In the Prescribed Calendars run, wheat sees a slight (though insignificant) improvement in its interannual variability (Fig. 8f), but a worsening of its overestimate of mean global yield (Fig. 5f) and production in most regions (Fig. S21f). Cotton and sugarcane see substantial improvements in global mean yield (Figs. 6b, e) 385 but mixed or worsened performance in terms of spatial pattern (Figs. 6b, f) and interannual variability (Figs. 8b, e). This may result from a number of factors, including incorrect planting area maps and management inputs, unrepresented management processes and/or varieties, inaccurate parameter values, and unrealistic physiology.

  The model's performance for global rice yield (Fig. 5), as well as regional rice yields in the tropics (Fig. 6), is much worse in the Prescribed Calendars run. However, the apparent better performance of the CLM Default calendars is only possible 390 because of high rates of yield failure in the subtropics and temperate zones (Fig. 7) due to incorrect model growing seasons. There is thus an argument for using observation-based growing seasons in the model to avoid such unrealistic failures, even if it degrades global yield performance.

  When reworking the growing seasons of rice or any crop, it would be important to reconsider the crop-specific physiological parameterizations used in CLM5. Many such parameters have a wide range of values in the literature, and some are not directly 395 observable. As such, crops in CLM have been parameterized within realistic bounds to produce realistic yields using what are



in some cases unrealistic growing seasons. Thus, notable changes to the growing seasons such as those seen in the Prescribed Calendars run—especially for rice, whose approximate doubling of global mean yield is due in large part to the elimination of excess yield failures in China—must be paired with a comprehensive re-evaluation of crop parameter values.

Moreover, simply using the prescribed sowing dates and maturity requirements used here would be too inflexible. A com-
bined approach, with observation-derived sowing and maturity dates as guidelines, may further improve simulations. For example, keeping CLM's sowing windows method but centering each gridcell's window on its GGCMI3 sowing date would allow interannual variation based on weather as well as some long-term shifts with climate change. Similarly, the algorithm in CLM5 that adjusts maturity requirements based on long-term climate history could start with the gridcell-level requirements used here instead of global values. In addition, comparing the GGCMI3 growing seasons with other datasets (e.g., Sacks et al.,
2010) would allow an assessment of the robustness of results seen here.

The sensitivity of yields to viability threshold (Sect. 3.2.4) suggests two possible strategies for model development and use. The Prescribed Calendars run sees substantially lower yields for corn, cotton, and rice when requiring 100% maturity instead of the GGCMI thresholds. In reality, if a farmer wanted to harvest at 100% maturity, they might leave their crops in the field for a few days longer; thus, CLM's maximum growing season length parameter might be artificially limiting yields. On the
other hand, crops should not be allowed to stay in the ground indefinitely. A more flexible "premature harvest" algorithm in CLM might better replicate real-world decisionmaking, perhaps only harvesting if the maximum season length is exceeded and no growing degree-days are accumulated for some number of days. Forecasts of a cooler-than-average season (i.e., requiring longer to reach maturity) might in real life induce earlier sowing, a behavior not present in CLM but whose effect on growing season length could be approximated using such a flexible algorithm. The simplest approach, however, would be to use a value
of less than 100% maturity as the viability threshold in postprocessing.

## 5   Conclusions

The development work described here enabled CLM to use externally-specified sowing dates and maturity requirements. CLM is thus now able to use sowing dates and maturity requirements generated by potentially more-realistic algorithms, without needing to code those into CLM itself. This also allows CLM's participation in global model intercomparisons that require the
use of standardized crop calendars.

While directly using observation-derived growing seasons for periods would be overly simplistic for prognostic simulations, doing so here has provided insights on areas for improvement in the built-in prognostic crop calendar functions and crop parameterizations generally. Cotton, rice, sugarcane, and wheat see large increases in global yield with these new inputs relative to the standard CLM setup. This represents a performance improvement for cotton and sugarcane but an overestimate for the
others. Global yield increases with the prescribed calendars are largely driven by production differences in a few small regions, some of which see yields further from observations than with standard CLM. That using more-realistic growing seasons sometimes decreases yield performance suggests that some crops may need to be reparameterized to function correctly when grown in a more realistic part of the year.



The growing season changes associated with the use of the prescribed calendars also result in a 15% reduction in global
irrigation demand across all cropland, driven mostly by rice and corn. This does not necessarily mean that real-world growing
seasons are optimized to minimize irrigation needs, as cotton, sugarcane, and wheat see slight increases in irrigation demand.
As with yield, the irrigation differences are driven by a few small regions for each crop, with South Asia playing an especially
large role. Because of the influence of land-atmosphere water fluxes on regional climate, climate simulations there may thus
be improved by the use of more growing seasons.

While this work was focused on the crop module in CLM, other models that use similar setups to determine sowing date
and/or maturity requirement might see similar effect sizes—and glean similar insights—from analyses such as demonstrated
here.

*Code and data availability.* Three versions of the CLM5 code were used for different parts of these experiments: Model spinup (https://doi.org/
10.5281/zenodo.7724294); 1850–1957 period and GDD-Generating run (https://doi.org/10.5281/zenodo.7724212); and CLM Default and
Prescribed Calendars, Sowing, and Maturity runs (https://doi.org/10.5281/zenodo.7724225).

Python code used for postprocessing GDD-Generating runs to produce prescribed sowing date and maturity requirement files, as well as
for general analysis and figure production, is archived at https://doi.org/10.5281/zenodo.7758123.

Prescribed calendar inputs and experimental outputs are archived at https://doi.org/10.5281/zenodo.7754247.

## Appendix A:  Crop calendars in CLM

### A1   Original formulation

Winter wheat was added to the crop module in CLM4.5 (Lu et al., 2017), but work to add an input mask or an algorithm telling
the model where to plant winter vs. spring wheat is still ongoing. Thus, only spring wheat is planted in these simulations, and
the description of growing seasons here and in 2.1 does not include the processes specific to winter wheat.

Planting occurs in the hemisphere-specific sowing window for a crop functional type (CFT) if and when three CFT-specific
thresholds are satisfied:

1. the 10-day running mean of temperature at 2 m ($T_{2m}$) is above the threshold $planttemp$,

2. the 10-day running mean of daily minimum $T_{2m}$ is above the threshold $minplanttemp$, and

3. the running 20-year mean of $GDD_8$ (a measure of total growing season warmth; see below) is greater than or equal to
   the threshold $gddmin$.

On the last day of the sowing window, the first two conditions are ignored, and the crop is planted as long as its long-term
average climate has been appropriate.





After sowing, at every timestep $T_{2m}$ is above a CFT-specific base temperature $T_{base}$, the crop accumulates

$$\min\left(T_{2m} - T_{base}, \Delta GDD_{max}\right) \times \frac{dtime}{D} \qquad (A1)$$

growing degree-days (GDDs; °C day), where $D$ is day length in seconds and $\Delta GDD_{max}$ is the maximum daily accumulation

allowed. This equation, with $T_{base}$ = 8°C and $GDD_{max}$ = 30 °C day, is also used during each year's $GDD_{Tb}$ period (April through September in the Northern Hemisphere, October through March in the Southern Hemisphere) to calculate $GDD_8$, the measure of total growing season warmth used in the sowing date calculation.

GDDs are accumulated in the heat unit index variable HUI, which is set to zero upon planting. The crop is harvested once HUI reaches the maturity threshold $GDD_{mat}$ or if the growing season has reached its CFT-specific maximum length. $GDD_{mat}$

is calculated differently for various crop groups, but is generally—within some minimum and/or maximum values—the 20-year running mean of $GDD_{Tb}$ (or some fraction thereof) for base temperature 0, 8, or 10°C. The base temperatures of spring wheat and sugarcane within 30°N and S vary based on latitude as $T'_{base,c} = T_{base} + 12 - 0.4|lat_c|$, where $T'_{base,c}$ is the adjusted base temperature for the CFT in cell $c$ with latitude $lat_c$.

In addition to maturity, other phenological stages are also defined relative to $GDD_{mat}$. The leaf emergence period begins

once the weighted-average accumulated GDDs of the top two soil layers (i.e., Eq. A1 but with $T_{soil}$ instead of $T_{2m}$) has reached, depending on CFT, 1% to 5% of $GDD_{mat}$. Grain fill begins once HUI has reached 40% to 65% of $GDD_{mat}$, again depending on CFT. Grain fill can also begin once the plant's leaf area index (LAI) has reached its CFT-specific maximum, in which case HUI is "boosted" to the CFT's threshold for the beginning of grain fill.

### A2   Enabling externally-prescribed growing season criteria

We allow gridded and optionally time-varying values of sowing date to be read from an input file, overriding sowing window and the related planting criteria. We have also enabled this functionality for $GDD_{mat}$. The choice to enable read-in $GDD_{mat}$ and not maturity date may at first seem peculiar, as the latter is more readily observable. However, it is important that CLM not harvest before physiological maturity, which is reached earlier in warm seasons and later in cool seasons. Specifying $GDD_{mat}$ allows this "floating" of harvest date.

When simulating crop cultivars whose $GDD_{mat}$ is known, it can be provided directly. However, in cases where only maturity date is known—as in this work, with the GGCMI3 dataset—an extra model run and some analysis must be performed. To generate maps of $GDD_{mat}$ for each crop, we must find the mean GDDs accumulated in each grid cell ($GDD_{accum}$, which is just HUI minus any "boosts" as described in Appendix A1) between the provided sowing and maturity dates over some reference period. Since the GGCMI3 protocol uses the 1980–2009 growing seasons as the reference period, we run CLM

from 1980–2010 (to allow for growing seasons begun in 2009 to complete the next year, if needed) with every crop remaining unharvested until the day before the next sowing. This run uses a slightly modified version of the CLM land use inputs to ensure that every crop a gridcell has at any timestep in the time series is included in every timestep.

All data from this "GDD-generating" run are discarded except for instantaneous daily values of $GDD_{accum}$. A postprocessing script, generate_gdds.py, then determines the mean value of $GDD_{accum}$ in each grid cell on the provided mean



maturity day and saves this as a new CLM input file, with a separate variable for each simulated crop PFT. This input file is then read by CLM as prescribed $GDD_{mat}$ values. To avoid unexpected behavior when very small values of $GDD_{mat}$ are in a denominator, we set a minimum of 1 °C day when reading in $GDD_{mat}$.

    Having CLM harvest on the GGCMI maturity date and save the GDDs accumulated between sowing and harvest would have been conceptually simpler but practically more complex, as it would have introduced more code that would need to be kept

up-to-date as model infrastructure evolved. The chosen method also allows GDDs to be re-generated without another model run if target mean maturity date changes in the future. Finally, it removes the possibility of a model user prescribing maturity date in a model run, which as discussed above would remove the needed ability of harvest timing to "float" with growing season temperature.

    Note that the GGCMI dataset provides two growing seasons for rice in some gridcells. Here, we use the growing season

associated with the largest area, according to the `fraction_of_harvested_area` variable included in the GGCMI growing season netCDF files.

    So far, `generate_gdds.py` and the other work presented here have only been used to generate static inputs of sowing date and $GDD_{mat}$. Future work will enable the production of time-varying inputs as well.

## Appendix B: Calculating yield

CLM saves a variable called `GRAINC_TO_FOOD` that is the basis of our yield calculations. However, some extra work is required for this variable to be comparable to what is reported in the observation datasets. We postprocess `GRAINC_TO_FOOD` as follows:

1. Assuming that 15% of yield is lost between the field and the wider food system, multiply by 0.85 ("harvest efficiency"; Lombardozzi et al., 2020).

2. Assuming that carbon is 45% of total harvested dry biomass, divide by 0.45 to get total dry biomass (Lombardozzi et al., 2020).

3. Divide by dry matter fraction (crop-specific, see Table B1) to get total harvested wet matter.

    Because the FAOSTAT-reported sugarcane yields include not just soluble solids and the water in which they are dissolved, but also a substantial amount of fiber biomass, an additional calculation is required. Specifically, we divide by 0.51, since

soluble solids (85% of which are sugars) represent only 51% of the solid matter in harvested sugarcane, with fiber the other 49% (Legendre, 1988).

    Note that Step 3 has not been performed in previous CLM evaluations, except for sugarcane. For this reason, reported CLM Default yields when including all harvests ("CLM Default (1)" in Fig. 9; see Sect. 2.4.1) are expected to be higher than in Lombardozzi et al. (2020) despite using the same run configuration and yield viability threshold.



**Table B1.** Dry matter fraction assumed for each crop.

| Crop | Dry matter fraction | Source |
|------|--------------------|--------|
| Corn | 0.88 | Wirsenius (2000), Table A1.II |
| Cotton | 0.912 | Wirsenius (2000), Table A1.III |
| Rice | 0.87 | Wirsenius (2000), Table A1.II |
| Soybean | 0.91 | Wirsenius (2000), Table A1.II |
| Sugarcane | 0.255 | Legendre (1988), Table 1 |
| Wheat | 0.88 | Wirsenius (2000), Table A1.II |

*Author contributions.* All authors contributed to the development of the research question and experimental design. SR wrote model code and performed the experiments with guidance from WS. SR performed analyses and composed the manuscript, with AR, LX, and DL helping with review and editing.

*Competing interests.* The authors declare no competing interests.

*Acknowledgements.* This work was supported by a gift from SilverLining's Safe Climate Research Initiative. Alan Robock and Lili Xia are supported by US National Science Foundation grants AGS-2017113 and ENG-2028541. The CESM project is supported primarily by the US National Science Foundation. This material is based upon work supported by the National Center for Atmospheric Research (NCAR), which is a major facility sponsored by the US National Science Foundation under Cooperative Agreement No. 1852977. Computing and data storage resources, including the Cheyenne supercomputer (doi:10.5065/D6RX99HX), were provided by the Computational and Information Systems Laboratory (CISL) at NCAR. NCAR is sponsored by the US National Science Foundation.

We thank Keith Oleson for assistance with model spinup, as well as Peter Lawrence and Samuel Levis for providing the CTSM 5.2 surface and land use input datasets.



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
