# Peer review of "Observation-based sowing dates and cultivars significantly affect yield and irrigation for some crops in the Community Land Model (CLM5)"

_Geoscientific Model Development, 2023_

## Author Comment (AC1)

We thank both reviewers for their kind comments as well as their suggestions, which have improved the paper.

Throughout this document, reviewer comments will appear in **bold** and our responses in plain text. New text in quoted passages will be indicated with underlines.

**Response to Reviewer 1**

**The aims of the presented study are to 1) incorporate externally-prescribed growing season data into the Community Land Model (CLM) and 2) use the modified model to explore the effects of the Global Gridded Crop Model Intercomparison (GGCMI) growing season data on historical yield and irrigation water demand simulations.**

**The research conducted is interesting and beneficial to the agricultural and modeling field. It sheds light on the new prescribed calendar functionality of the CLM model, but also shows limitations of the simulated global phenology results. The methodology is well described, and the conclusions are well derived. However, there are some minor issues that hinder the clarity of the study. Minor revision is recommended before acceptance.**

**General comments:**

1. **The study often refers to wheat in general (e.g., in text and figures 9 and 10 labels), but for CLM5, wheat is only represented by spring wheat. The authors mention this several times but there are still areas which may suggest that winter wheat is considered in the analysis. I also find it difficult to draw conclusions for global wheat phenology when vernalization is not considered since it plays a key role in determining planting date for wheat farmers and the length of the growing season. Further clarification should be added throughout the manuscript to ensure readers understand that only spring wheat is considered.**

We have specified "spring wheat" in various places that previously read only "wheat": lines 10, 75, 262, 301, 302–303, 350, 356, 456, 464, and Fig. 6 caption.

We have also added text in various places to note where the use of spring wheat everywhere complicates analysis or future work:

- Lines 252–254: "A 0.6 t ha-1 (21%) overestimate for wheat in CLM Default is increased in Prescribed Calendars to 1 t ha-1 (50%), although interpretation of this is complicated by the use of spring wheat everywhere."
- Lines 293–294: "Wheat's correlation increases slightly and corn's decreases slightly, but both remain significant at the $p < 0.05$ level—although, again, interpretation for wheat is complicated by the use of spring wheat everywhere."

2. **What is the reason for the huge difference in sugarcane shown in S15? Most areas are > 200 days different which is an entirely different season.**

Sugarcane has previously not received much attention in CLM, and it is unclear exactly where its seasonality parameterization comes from. It is thus not entirely surprising that CLM Default growing season length is so wrong, but it is notable that the Prescribed Calendars map still differs so much from GGCMI. This likely stems, in large part, from the CLM-imposed maximum growing season length of 300 days. Since most of the sugarcane area in the GGCMI dataset has a value higher than that (Fig. S15a–b), even the Prescribed Calendars setup starts with a disadvantage. In addition, Prescribed Calendars uses the *average* accumulated growing degree-days during the GGCMI-derived growing season, limited to 300 days after sowing across most of sugarcane's planted area. Assuming a normal distribution, approximately 50% of seasons will need more than 300 days to accumulate sufficient growing degree-days, in which case they will be harvested upon reaching the maximum season length at day 300. This will further reduce the mean growing season length, as visualized here for 1000 samples from a normal distribution with mean 300 days and (arbitrary) standard deviation 30 days:

[Figure]

**2. [continued]: Why even include sugarcane and cotton if they are not used in most global studies due to limited confidence (as stated in lines 304-308)? It is challenging to have confidence in the simulated results with this large of a difference. A detailed section on model uncertainties should be included in the Discussion.**

While the unique growth forms of sugarcane and cotton are a large part of why they're not typically included in global models (as you note we stated on lines 304–308 of the original manuscript), they are indeed included in out-of-the-box CLM runs. We thus consider it important to include them here—any improvement in realism is valuable, especially when it's as large as seen for these crops.

The reviewer raises important questions about model applicability. *Should* sugarcane and cotton be simulated in most CLM runs? When they are included, how much trust should we have in the results? Unfortunately, answering these questions requires a discussion of applicability and

fitness-for-purpose philosophies that be more appropriate for a workshop than a subsection of our discussion. These questions are thus outside the scope of this work, which is focused on understanding how to improve the model we have.

3. **The median of the prescribed calendar GDD for cotton are much higher than the CLM default in Fig. S17. This results in the large yield increases seen, but how are these GDD values justified? Additionally, the prescribed calendar GDD variation for all crops is unrealistic (e.g., < 500 or > 3000 GDD, > 5000 GDD for sugarcane), and should be checked. How are these simulated results justified? Improved justification and uncertainty explanation is needed in the Discussion to have confidence in the model results.**

There are two main reasons that the maturity requirements from the GDD-Generating run and postprocessing may be unrealistically low or high.

● The GGCMI growing season dataset cannot be assumed to be 100% accurate across all regions for the entire time period. A certain amount of extrapolation was required to achieve global coverage, and even in places with source data, it may represent only a portion of the 1980–2009 period. GGCMI growing seasons that are longer than realistic would produce too-high maturity requirement values, and vice versa.
● The crop-specific base temperatures (the temperature below which no GDDs accumulate) in CLM may be too low, which would cause unrealistically high maturity requirements (and vice versa).

This second point is critical. Real-world cultivars of a given crop can vary widely in their base temperature, but CLM uses globally-constant values for cotton, rice, and soybean. Corn's base temperature takes one of two values: 8°C for the temperate PFT and 10°C for the tropical one. Some more variation is possible in CLM's spring wheat and sugarcane, whose base temperatures vary linearly from the Equator to 30 degrees absolute latitude (being held constant beyond that). Spring wheat's base temperature goes from 0°C at the Equator to 12°C at and above 30 degrees absolute latitude, while sugarcane's ranges 10–22°C.

The Prescribed Calendars setup described in this study is intended only for extremely limited applications; specifically:

● evaluating how much of a difference realistic growing seasons could make to yield and irrigation requirements, as in this study; and
● conforming to model intercomparison protocols such as GGCMI phase 3.

As alluded to in the Discussion, using the sowing dates and maturity requirements from this study in an actual application of CLM—even, to some extent, an intercomparison—without any reparameterization would be unwise:

> Many such parameters have a wide range of values in the literature, and some are not directly observable. As such, crops in CLM have been parameterized within realistic bounds to produce realistic yields using what are in some cases unrealistic growing seasons. Thus, notable changes to the growing seasons such as those seen in the Prescribed Calendars run—especially for rice, whose

approximate doubling of global mean yield is due in large part to the elimination of excess yield failures in China—must be paired with a comprehensive re-evaluation of crop parameter values.

Reparameterizing for more realistic base temperatures (parameter `baset` and, for sugarcane and spring wheat, `baset_latvary_slope` and `baset_latvary_intercept`) might produce more realistic maturity requirements without much effect on growing season or yield. Other parameters related to crop development may have more leverage:

- the GDD accumulation values, expressed as a fraction of maturity requirement, at which crops transition to the vegetative (`lfemerg`) and reproductive (`grnfill`) stages;
- the maximum number of GDDs that may be accumulated in any one day (`mxtmp`); and
- maximum growing season length (`mxmat`).

In addition, there are a number of non-crop-specific parameters that may need to be adjusted, such as those controlling allocation of photosynthate to different parts of the plant.

We have added the above discussion to the text, as well as rearranged some existing text, into the new Sect. 4.3 (lines 427–455).

We have also added the following text in other places to make it clear that more development work would need to occur before using this setup in production runs:

- Lines 65–70: "Here, we have given CLM the capability to use externally-prescribed growing season data such as that required by GGCMI, enabling for the first time its participation in this important community effort.The specific setup introduced here is intended only for limited application: The prescribed values we use are too rigid for prognostic runs, and even with added flexibility, a comprehensive re-evaluation of parameters related to crop growth and development is warranted (Sect. 4.3). However, it does allow us to explore the effects of forcing CLM with the GGCMI growing season data on simulations of historical yield and irrigation water demand, and from those results to highlight where and how CLM performance could be improved."
- Lines 405–408: "Similarly, the algorithm in CLM5 that adjusts maturity requirements based on long-term climate history could start with the gridcell-level requirements used here instead of global values. Care would need to be taken to ensure that the resulting maturity requirements are within observed ranges, perhaps allowing for an increased maximum value in the future with genetic improvements."

**Specific comments:**

4. **Line 25-26: "…can reduce yield because they have less time to photosynthesize." Yield reduction from temperature driven early maturity is mainly because of less time for grain filling/reproductive growth and resource partitioning. Please clarify.**

We have changed the end of this sentence to read, "can reduce yield because they have less time to photosynthesize and allocate photosynthate to grain" (lines 25–26).

5. **Line 100: what is the resolution of the simulations?**

We have moved the explanation of our resolution from Sect. 2.3 to lines 103–104: "We regrid the half-degree data from the GGCMI crop calendar dataset to the approximately 2-degree resolution of our simulations (2.5° longitude × ~1.9° latitude, or the f19_g17 CLM grid) using…"

6. **Line 239 to 240: Remove parenthesis around sentence.**

The parentheses here indicate to the reader that the enclosed sentence is an aside intended to clarify "at least in terms of 'thermal time'" in the previous sentence, rather than a statement connecting that sentence to the last sentence of the paragraph. Without the parentheses, the end of the paragraph becomes more difficult to understand.

Ideally, the sentence in parentheses (lines 244–245 in revised manuscript) would instead be a footnote, but the *GMD* guidelines state that "footnotes should be avoided in the text, as they tend to disrupt the flow of the text." If the editor and/or copy-editor feel that a footnote would be more appropriate here, we'd be happy to make that change.

**Technical Corrections:**

7. **Line 37: Please also reference the Agricultural Model Intercomparison and Improvement Project (AgMIP, https://agmip.org/) when first mentioning GGCMI since GGCMI is a part of this larger project.**

The sentence now reads, "Global crop models, such as those participating in the Global Gridded Crop Model Intercomparison (GGCMI; Jägermeyr et al., 2021b) and its parent Agricultural Model Improvement Project (AgMIP; Rosenzweig et al., 2013), are…" (lines 38–39).

**Response to Reviewer 2**

**Rabin and colleagues have developed a novel function that enhances the capabilities of the CLM crop module, allowing it to incorporate externally provided planting and maturity inputs. This approach holds the promise of improved accuracy and flexibility compared to the default method embedded within the CLM crop module. Moreover, the newfound ability to utilize externally sourced growing season data marks a significant advancement, potentially paving the way for the integration of CLM into the Global Gridded Crop Model Intercomparison (GGCMI). This method bears particular importance for crop models exhibiting promising potential. The authors have executed an admirable job in presenting their findings. Nevertheless, I would like to offer a few minor suggestions:**

**The validation of the model involved a comparison of simulated crop yields (approximately 2-degree resolution), against data from the FAOSTAT database and the EarthStat dataset on country and global scales. However, this coarse resolution introduces uncertainties.**

**Could you please elaborate if you have conducted validation at a finer site-specific level? This would provide a more precise depiction of the new model's enhancements.**

We did not conduct any finer-scale evaluation of performance. In general, since CLM is a global model, we do not necessarily expect performance at any given site to closely correspond to field observations without the use of site-specific inputs (e.g., meteorological forcing data, soil characteristics, fertilizer amounts and timing) and CFT parameters updated to match the variety used at the site. This localization would include site-specific sowing date and maturity requirement (and/or harvest date), which of course would negate the work demonstrated here.

The non-crop portion of CLM has performed favorably when compared to flux towers (e.g., Lombardozzi et al., 2023, in press). However, there are limited flux tower data available at agricultural sites to perform a similar analysis, in addition to the need for crop-specific localization as described above.

**In reference to line 426, the statement "The use of more realistic growing seasons occasionally leading to decreased yield performance" suggests that certain crops may require reparameterization to accurately function during more realistic periods of the year. Could you kindly provide further details regarding the parameters employed in the new model? Additionally, was there any calibration undertaken? This information appears somewhat unclear. Furthermore, it might be beneficial to include a section that highlights key parameters and presents a sensitivity analysis, elucidating the pivotal parameters in the new algorithm.**

We apologize for any confusion: Aside from the use of prescribed sowing dates and maturity requirements, the Prescribed Calendars run is identical to the CLM Default run (and, generally, CLM out-of-the-box setup). No new parameters were introduced, and no existing parameters had their values changed.

We have added a Discussion section to clarify this and to further explore what we mean by reparameterization; please see response to Reviewer 1 #3 above.

**On line 240, the parenthesis might be a typo.**
Please see response to Reviewer 1 #6 above.

**Other changes**

- Updated correspondence email from sam.rabin@gmail.com to samrabin@ucar.edu.
- Line 237: "Fig." corrected to "Figs."
- Lines 480–481: "Thus, only spring wheat is planted in these simulations, and the description of growing seasons here and in Sect. 2.1 does not include the processes specific to winter wheat."
- Lines 492–494: "where D is day length in seconds and ΔGDDmax is the maximum daily accumulation allowed (26 °C day for spring wheat and 30 °C day for all other crops). This equation, with Tbase = 8°C and ΔGDDmax = 30 °C day, is…"

**Works cited**

Lombardozzi, D. L., Wieder, W. R., Sobhani, N., Bonan, G. B., Durden, D., Lenz, D., SanClements, M., Weintraub-Leff, S., Ayres, E., Florian, C. R., Dahlin, K., Kumar, S., Swann, A. L. S., Zarakas, C., Vardeman, C., and Pascucci, V.: Overcoming barriers to enable convergence research by integrating ecological and climate sciences: The NCAR-NEON system Version 1, EGUsphere [preprint; in press], https://doi.org/10.5194/egusphere-2023-271, 2023.